# Exploring the Genetic and Functional Diversity of *Porphyromonas gingivalis* Survival Factor RagAB

**DOI:** 10.3390/ijms26031073

**Published:** 2025-01-26

**Authors:** Pauline G. Montz, Evdokia Dafni, Bernd Neumann, Dongmei Deng, Mohamed M. H. Abdelbary, Georg Conrads

**Affiliations:** 1Division of Oral Microbiology and Immunology, Department of Operative Dentistry, Periodontology and Preventive Dentistry, Rheinisch-Westfälische Technische Hochschule (RWTH) University Hospital, 52074 Aachen, Germany; pauline.montz@rwth-aachen.de (P.G.M.); evdokia.dafni@rwth-aachen.de (E.D.); abdelbarym@rki.de (M.M.H.A.); 2Institute of Clinical Microbiology, Infectious Diseases and Infection Control, Paracelsus Medical University, Klinikum Nürnberg, 90419 Nürnberg, Germany; bernd.neumann@klinikum-nuernberg.de; 3Department of Preventive Dentistry, Academic Centre for Dentistry Amsterdam (ACTA), University of Amsterdam and VU University Amsterdam, 1081LA Amsterdam, The Netherlands; d.deng@acta.nl

**Keywords:** *Porphyromonas gingivalis*, phylogeny, *ragAB* locus, RagAB transporter, PCR, genome mapping

## Abstract

*Porphyromonas gingivalis* is a key pathogen in periodontitis. Its outer membrane contains the RagAB transport complex, which has been implicated in protein uptake, essential for a proteolytic species. RagA is a 22-stranded β-barrel, and RagB is the corresponding 4-TPR lid, together forming a TonB-dependent system acting as a “pedal bin”. Four different alleles were observed, of which *ragAB-1* is more virulent than the others. Our aim was to map *ragAB* in 129 strains of *P. gingivalis* and related species available in our collection, supported by a newly introduced universal PCR for amplification/sequencing of all four *ragA* variants and to find reasons for the differences in virulence and/or fitness. Regarding the PCR method, by pairing established Long-PCR primers with our newly designed sequencing primers (ragA-F0, -F1, -R2, -R2a, -R4), it was possible to amplify and sequence all four *ragA* variants. The same was not possible for *ragB* due to high heterogeneity. The mapping allowed us to type all strains into *ragAB-1-4*. For each type, some strains (of mainly animal origin such as *Porphyromonas gulae*) with slightly different amino acid sequences were identified (designated *ragAB-1a* to *-4a*). In terms of function, the transfer of recently discovered SusCD information to the similar RagAB complex provided new insights. Substrate specificity as well as length of pedal could be the route to differential virulence (survival rate, fitness) as Rag-1 (closer related to Rag-3/4) and Rag-2 were found to be massively different here. In general, substrate–ligand-binding sites seem to be quite variable with the exception of Rag-1, probably indicating nutritional preferences. In addition, an insertion (8 aa long) found in loop L7 throughout RagA-2 could not only affect the dynamics of lid opening/closing but might also alter the associated substrate throughput rate.

## 1. Introduction

*Porphyromonas gingivalis* (*P. gingivalis*) is a key pathogen in severe forms of periodontitis, which is described as a chronic inflammatory disease of the teeth-surrounding tissues based on robust dysbiosis. *P. gingivalis* is an obligate anaerobic, Gram-negative, rod-shaped to pleomorphic, immobile, black-pigmented, and asaccharolytic/proteolytic bacterium [1,2]. Together with *Treponema denticola* and *Tannerella forsythia*, it forms the so-called “red complex” [3], which is partly responsible for the induction of inflammation, bleeding on probing, the formation of deep gingiva pockets, attachment loss, and, ultimately, the destruction of the periodontium. *P. gingivalis* has been linked with a number of systemic diseases, including Alzheimer’s disease [4,5], rheumatoid arthritis [6], vascular malfunctions [4], and preterm birth [7], as reviewed elsewhere [1]. The intensive study of its virulence factors, including capsule, outer-membrane vesicles, lipopolysaccharide, gingipains, and fimbriae, recently shown for long fimbriae by our group [8], has not yet yielded a comprehensive explanation of the differences in pathogenicity or fitness observed between different strains, which is why further genetic investigation of *P. gingivalis* is essential.

Receptor antigens RagA and RagB are major components of the outer membrane of *P. gingivalis* and have been linked with virulence because of their contribution to subcutaneous lesion development, epithelial cell invasion, and efficient growth of the pathogen by acting as a transport system for nutrients [9]. The RagAB importance has been confirmed in a murine model, as evidenced by the reduced virulence observed in a *ragB* mutant—and especially, a *ragA* mutant—in terms of murine lethality and lesion size [10]. Earlier studies have predicted that only some “more virulent” *P. gingivalis* strains such as W50, W83, or WPH35 possess the *ragAB* locus [11,12]. The lack of universal primers capable of amplifying all variants may contribute to this observation, as the *ragAB* locus (also abbreviated as *rag* and first described in 1999 [13]) exhibits considerable heterogeneity. Sequence variations of up to 30% in *ragA* and as high as 48–47% in *ragB* were observed [14]. Later studies have described four different variants of the *rag* locus, both in oral samples and in *P. gingivalis* strains, named *rag-1* to *rag-4,* of which type *rag-1* (including W50 or W83) appears to be more virulent [15,16,17]. Genes *ragA* and *ragB* are co-transcribed [13], and their expression is highly interdependent, in the sense that a *ragA* mutant significantly reduces the expression of RagB, and a *ragB* mutant leads to degraded RagA [14]. The *ragA* operon (3.1 kb) results in the production of RagA (PG_0185, GenBank: AAQ65420.1), a 115 kDa TonB-dependent 22-stranded β-barrel transporter (TBDT) with a plug domain blocking the substrate passage through the outer membrane [16,18]. RagA is covered by the 55 kDa substrate-binding lipoprotein RagB (PG_0186, GenBank: AAQ65421.1), encoded by the *ragB* operon (1.5 kb) 30 bp downstream [9,12,14,18]. Crystallography revealed the RagAB structure of strain W83 as a heterotetrameric complex (RagA_1_B_1_-RagA_2_B_2_ dimer) acting as a double “pedal bin” with monomers opening independently, enabling the uptake of the gingipain-cleaved proteinaceous nutrients/peptides [18,19]. The binding of an extracellular substrate (peptide) leads to a conformational change making it accessible for interaction with TonB, an inner membrane protein complex. This interaction allows the formation of a substrate transport channel in the periplasmic space [18]. The fact that the RagAB complex is similar to other TonB-dependent outer membrane transport systems, such as SusCD, which is involved in malto-oligosaccharide and starch uptake in *Bacteroides thetaiotaomicron* [20], will be used to critically transfer structural and functional knowledge. While other virulence factors of *P. gingivalis* have been extensively studied, the genetic and functional diversity of RagAB remains poorly understood.

The objectives of this study were the following: first, to demonstrate the presence of the *rag* locus in a maximum number of strains and to classify them accordingly; second, to investigate the genetic differences between the four *rag* types; third, to compare functional domains of the deduced amino acid sequence; and finally, to develop a universal PCR method capable of amplifying all *rag* variants.

## 2. Results

### 2.1. Developing a Universal RagA-Gene-Directed PCR Method

We begin by providing a brief overview of the achievements derived from the universal *ragA*-directed PCR. Attempts to develop a universal (type independent) *ragB* amplification were unsuccessful due to its high degree of heterogeneity. However, if *ragA* type is determined, the corresponding *ragB* can be amplified by type-specific primers as published by Hall et al. in 2005 [16]. Subsequently, we present a comprehensive summary of the mapping results.

Amplification of the *ragA* gene (for details and Figures see Section 4): We were able to sequence the *ragA* locus using the Long-PCR product as a template and the ragA-primers for sequencing. The LPCR primers of Hall et al. (2005) were able to amplify 6–8.5 kb products, which were visualized via agarose gel electrophoresis [16]. Type-1 led to the biggest product with approximately 8.5 kb (because of IS*Pg* insertion and a long *ragB*-PG0188 spacer encoding a hypothetical peptide ID SEQF1064.1_00180), followed by type-4 of almost 8 kb (with an extra-long *ragB*-PG0188 spacer encoding the C-terminal domain of Arg- and Lys-gingipain proteinase, ID BAG32814.1). In contrast, type-2 and -3 LPCR products were notably shorter with 6 kb. Regarding sequencing results with the double cleaned LPCR product as template, type-1 and -2 strains produced ambiguity-free *ragA*-sequences with only about 40–50 bps missing at both ends. In contrast, for type-3 strains and primer ragA-R2 an ambiguity-rich sequence occurred, but this problem could be solved by applying Ripseq (Pathogenomix, Santa Cruz, CA, USA), an online platform which is able to resolve overlapping sequences [21], and where reference sequences of *ragA*-1-4 (W83-SEQF1064|AE015924.1, LyEC01-CP126309.1:280912-284052, KCOM 2796-CP024597.1:1253375-1256473, ATCC33277-SEQF1538|AP009380.1) were deposed and can be accessed on request. As primer ragA-R2 might fail for sequencing type *ragA*-4, a new primer (ragA-R2a) was designed which produced an almost ambiguity-free sequence. However, in the case of ambiguities, Ripseq can again be helpful in resolving them. As *ragA* and *ragB* types are always correlated—so far known and confirmed in our study—the universal *ragA* amplification/sequencing can also be used to predict the *ragB* type and select the corresponding type-directed sequencing primer published by Hall et al. (2005) [16].

### 2.2. Genome Mapping Results

First, we confirmed that all 129 *P. gingivalis* strains investigated contained the *ragA* and *ragB* genes. Additionally, *ragA* and *ragB* types were corresponding in every genome and were apparently transmitted in a dependent manner.

For *ragA*, interestingly, the first 450 bases (150 aa) were nearly identical for all four types and thus very conserved and type-independently essential. Contrarily, *ragB* was very heterogeneous throughout the entire gene. Appendix A gives an overview of typing results of our isolates together with strain information (synonymous strain numbers, place and date of isolation, host if not human). Clearly, *ragAB*-2 was the most abundant with almost 50% (64 strains), followed by *ragAB*-1 with 17% (22 strains, confirming earlier PCR-based results [22], *ragAB*-3 with 17%, and *ragAB*-4 with 16% (21 strains). It is notable that isolates from Germany (cities of Aachen, Freiburg, Kiel, Mainz, and Nürnberg) were over-represented in our strain collection, which could introduce a potential bias.

### 2.3. Phylogenetic Tree Building

After importing all *rag* sequence data (most as a result of genome mapping, a few as a result of PCR amplification and sequencing) into MEGA11, the neighbor-joining phylogenetic trees with 100 bootstrap replications on the DNA-level, shown in Appendix A, as well as—after in silico translation—on the protein level, shown in Appendix A, were created. Four clusters could be observed for genotypes as well as protein-/serotypes according to the four different *ragAB*/RagAB alleles. In every type, however, a few strains were identified with a slightly different aa sequence (based on a significantly different bp sequence) which we refer to as type-1a to type-4a. For instance, strains OMI 1072 and 1081 (both *P. gulae* isolated from monkeys) demonstrated a variant of *rag*/Rag-1 (designated *rag*/Rag-1a), OMI 1119 (synonymous no. HG3637, JKG6; matching also with HG1691 and LyEC01/02) a variant of *rag*/Rag-2 (designated *rag*/Rag-2a), OMI 1076 and 1128 (both *P. gulae* isolated from cat and dog) a variant of *rag*/Rag-3 (designated *rag*/Rag-3a), and—finally—OMI 1080 and 1160 (again *P. gulae*) a variant of *rag*/Rag-4 (designated *rag*/Rag-4a). Next, we performed a type-specific analysis.

Type-1: Cluster *ragAB-1* included two out of five investigated *P. gulae* isolates (OMI 1072, OMI 1081, both from monkeys) that were slightly different by sequence (*ragAB-1a*) as outlined above and 20 human isolates. No geographical pattern was observed, as human isolates were from Europe, America, and Asia. Four strain pairs of Indonesian people that contain the same *ragAB* type during the span of eight years fall in this cluster (1994/2002: OMI 1079/1101, OMI 1084/1108, OMI 1078/1127, OMI 1087/1125). However, one Indonesian patient showed type-1 in 1994 (OMI 1068) but type-4 in 2002 (OMI 1062-2). This was confirmed by whole-genome comparison where OMI 1068 and 1062-2 had a 0.08 PhyloPhlAn-calculated distance while pairing strains were (almost) identical.

Type-2: Cluster *ragAB*-2 showed one minor outlier (OMI 1119 synonymous JKG6). Otherwise, we observed a very high similarity among the remaining 63 isolates which can only partially be explained by the fact that many of these strains came from Germany. Seven pairs of isolates from Indonesia belonged to *ragAB*-2. No geographical pattern was observed, as isolates were from Europe, America, Asia, and Africa. Notably, OMI 731 could not be included at first because of gaps in the mapping sequences, but these could later be edited using the PCR method described.

Type-3: Cluster *ragAB*-3 contained two *P. gulae* strains, OMI 1076 (from cat) and OMI 1128 (from dog) that were phylogenetically apart from the other isolates (variant *rag-3a*). In general, type-3 was more heterogeneous compared to type-1/2. Type-3 included isolates from Germany, Belgium, The Netherlands, Sweden, USA, and one pair of isolates (1994/2002: OMI 1088/OMI 1057) from the same Indonesian patient. The sequences between the related isolates, however, were distinct, which was confirmed on the whole-genome level. Thus, this Indonesian patient was either colonized by two different *P. gingivalis* strains or switched strains with the same *rag* type between 1994 and 2002.

Type-4: Cluster *ragAB-4* included two proven *P. gulae* strains phylogenetically apart from the other isolates of type-4 (variant *rag-4a*). Type-4 included isolates from Germany, Japan, USA, Canada, and Indonesia. Among the Indonesian patients, two matching pairs were isolated with eight years of distance (1994/2002: OMI 1046/OMI 1104 and OMI 1054/OMI 1105). An additional Indonesian patient presented a *ragAB* type-1 isolate in 1994 (OMI 1068) but a type-4 isolate in 2002 (OMI 1062-2), both also not closely related on the whole-genome level. Besides the *P. gulae rag-4a*, at least one additional subcluster (*rag-4b*) was recognized.

Since *ragAB* type-1, the most virulent, was identified in monkeys, we propose here that it may represent the ancestral sequence of the *ragAB* gene cluster. However, according to our phylogenetic trees, several other animal lineages exist, and horizontal transfer (HGT) of the *ragAB* gene cluster between *P. gulae* and *P. gingivalis* may be possible on several independent occasions and may even be ongoing, as is known for other virulence genes in *P. gingivalis/P. gulae*, such as the *fim* genes [8].

We were also interested in any evidence of a possible separation/break of *ragA* and *ragB* co-evolution/transfer. In general, with little variation in tree topology, the assignment to the four different types remained the same (see Appendix A). As a very minor exception of the strict co-evolution, strains OMI 1110, OMI 884, OMI 1126-14, OMI 1107, OMI 1091-1, and OMI 1085 formed different subclusters comparing *ragB-2* with *ragA-2*. Whilst for *ragA* type-1 and -2 were grouping together, on all other levels (RagA, *ragB*/RagB) type-1, -3, and -4 were grouping together and were in contrast to type-2 (Appendix A).

Our next step was to analyze the extent to which the amino acid sequence differences had an effect on RagAB presumable function and ultimately on *P. gingivalis* virulence. By comparing RagAB with SusCD, where a substantial amount of information regarding functional domains is available, and by integrating the amino acid sequence data of 129 OMI strains with 100 top matching NCBI data, we were able to gain further insights into the essential structures and functions of the barrel and lid. Nevertheless, while SusCD is responsible for glycan uptake, RagAB-1-4 is responsible for peptide uptake, a distinction that must be acknowledged.

### 2.4. Analysis of RagA Functional Domains

First, a neighbor-joining phylogenetic tree based on the top 100 aligned and currently (date: 31 October 2024, source NCBI) available protein BLAST RagA sequences of *P. gingivalis* and related species was created (Appendix A). As some sequences, that are identical but observed in different strains, have been grouped into the same non-redundant RefSeq WP numbers, a translation into associated strain numbers is provided with Appendix A (“WP-strain numbers translation”). For RagA, the percentage identity of the W83 blastp results reached from ~97–96% (type-1) over ~69–67% (type-4 and -3) to ~64–58% (type-2). Even the outgroup had a percentage identity of ~53–49% to the reference W83.

Second, a maximum alignment, integrating NCBI and OMI RagA sequences, was presented (Appendix A). The objectives were as follows: i. to identify areas that might explain the different virulence/fitness of the four types, ii. to further search for conserved domains representing structures of fundamental importance, and iii. to exclude highly variable regions as less important. Relevant domains (based on studies of Madej et al. 2020 and White et al. 2023 [18,23]) were therefore viewed from several angles: similarity to the reference W83 (thus to Rag-1, reference color blue), hydropathy (reference color blue = water soluble), RasMol (grouped by similar aa properties), and size of aa side chains (reference color blue = long).

The TonB box (X103-G108) (varieties shown in Appendix A left) is located inside the barrel and is shifting in position between an open and closed state [18]. In fact, among the four RagA types, we did not find any difference at the protein level which confirms its immense importance (see homogenous colors over all strains, types, and aa features). Next, as part of the plug domain, region X211-A219 (Appendix A right) represents an important area going through conformational changes during the process of substrate binding and passage [18]. It is interesting to note that, while otherwise conserved among *P. gingivalis*/*P. gulae*, serine (S211, polar, hydrophilic) instead of alanine (A211, non-polar, hydrophobic) is found here for type-2 and outgroups including *Bacteroides thetaiotaomicron*. Given the crucial role of the plug in facilitating substrate passage, this single position and its change in polarity may have played a pivotal role in the evolutionary trajectory towards a more (or contrarily less) virulent/fit phenotype. In contrast, the RagA loops that are positioned on the outside of the membrane show more sequence variability.

In total, 11 loops can be counted, of which loops L7 and L8 were assigned a major role by Madej et al. [18]. We further propose that loop L7 functions as a genuine “pedal” capable of raising the RagB lid while loop L8 is holding the RagB lid to limit/secure its movement. Figure 1, modified from iCn3D, shows the RagAB transport dimer in the open/closed state. Loops L7 and L8 are highlighted, showing the stable association with RagB. For loop L7, depicted in Appendix A (left), there is an insert of usually eight amino acids (GNPEYYAH) in RagA-2 with a few sub-types (as in COT-052 *P. gulae* strains from Leicestershire, isolated 2012). Transferring this structural peculiarity to its role as “pedal”, a possibly larger opening of the lid could have a crucial influence on the absorption of the substrate and on the lid opening/closing dynamics. Loop 8 is not only involved in holding the lid but also in substrate transport. Appendix A (right) shows L8 region X729-X753 to document the many differences among RagA -types, with type-4 showing both inter- and intra-type variations. Known from the literature (SusCD as model [23]), substrate binding leads to crucial conformational changes: W685 (on the base of loop 8, highly conserved in SusC and RagA-1-4) shifts inwards and pushes F583 up (conserved) and S193 down (polar uncharged serine in SusC is changed to alanine in RagA). As outlined above, without any ligand (apo), amino acid Y191 (conserved), which is part of the same plug loop, forms a triple aromatic stack with Y89 (nearby the TonB box, conserved) and F558 (barrel wall, conserved) that links the barrel wall, the TonB box, and the plug domain together (Figure 1). If a substrate is bound, Y191 shifts towards the periplasm, the aromatic lock is resolved, and the N-terminus is released, making the TonB box (energy transducer) able to interact with TonB, disrupting the plug and opening a channel into the periplasm [23]. Since we can transfer almost all the SusCD amino acids involved to the RagAB transport system, we can assume that the triple aromatic stack works in the same way here, resulting in an equal mechanism for plug triggering. Since these regions/positions are identical for SusC as well as for type RagA-1-4, we confirm their fundamental importance for the functionality of the nutrient acquisition protein complexes but conclude that they cannot explain the differences in virulence between the four heterogeneous RagAB types.

Next, we show potential RagA domains in contact with the proteinaceous substrate during uptake (Appendix A for multi-alignment and Appendix A, partA, for RagA-1 W83 in a three-dimensional structure). The most virulent RagA-1 possesses an alanine at X399 (instead of polar serine for RagA-2-4 and SusC) and methionine at position X797 (instead of valine for RagA-2-4 and SusC, source: NCBI). Methionine in proteins fulfils an important antioxidant role, stabilizes the structure of proteins, participates in the sequence-independent recognition of protein surfaces, and can act as a regulatory switch through reversible redox reactions.

Furthermore, while X400-RagA-1 is aspartic acid (negatively charged, polar), X400-RagA-2 is glycine, and X400 in both RagA-3 and RagA-4 is a polar uncharged serine. Additionally, X405-RagA-1/A-2 is identified as tyrosine and alanine (both hydrophobic) versus polar and uncharged asparagine for RagA-3/A-4. At X408, RagA-3/A-4 is again presenting asparagine in contrast to glycine for RagA-1 and serine for RagA-2. Interestingly, the region X399-X409 in RagA-1 has four aromatic side chains (1xF, 3xY) and type-3 and -4 have three (1xF, 2xY), while type-2 has only two (1xF, 1xY). SusC, on the other hand, has none. If we consider that aromatic compounds like to assemble in stacks, importance for the overall structure should be considered. However, the spatial distance makes participation in the triple aromatic stack unlikely. While N800 and T803 are the same for RagA-1, RagA-4, and RagA-3, the type for RagA-2 differs with glycine (G800) and tyrosine (Y803) and is markedly different, as it is less hydrophobic (less red in Appendix A, category hydropathy) and with a different pattern of side chains (less purple, more red or blue). Finally, and in contrast to all other substrate-binding regions, N893-Y898 —absent in SusC—is conserved among all RagA-1-4 types and thus excluded for variant or virulence definition but is probably instead important for switching from sugar to protein substrate.

While the aforementioned potential substrate-binding sites exclusively refer to proteinaceous ligands, it is necessary to screen for the fructo-oligosaccharide contact of SusC by F649, as proposed by White et al. [23]. Indeed, this region was found to be absent in all RagA.

Concluding here, the very conserved regions like the TonB box, the plug region (except position S211/A211), or the substrate-binding site N893-F898 can be excluded to explain the differences in virulence, but X399-X409, X799-X804, and the length of Loop 7 as the most expanding hinge (pedal) could be of interest here. Obviously, the virulent RagA-1 differs from RagA-2 with RagA-3/A-4 as possible intermediates. Figure 2 summarizes the peptide transport by RagAB.

### 2.5. Analysis of RagB Functional Domains

Like for RagA, a RagB phylogenetic tree based on the top 100 non-redundant RefSeq WP sequences was calculated (Appendix A), combined with all OMI-RagB sequences, and aligned by the NCBI Multiple Sequence Viewer 1.25.0 (Appendix A). In contrast to RagA, RagB was less conserved over the entire protein. The percentage identity, with W83 as reference, showed greater leaps from ~100–99% (type-1), ~57% (type-3), ~49–48% (type-4), ~48–47% (type-2), to finally ~33–27% (outgroups). While for RagA the outgroup included mainly *P. uenonis*, for RagB the outgroup contained a variety of *Porphyromonas* species (*P. asaccharolytica, P. endodontalis, P. gingivicanis, P. uenonis*).

Some relevant structures of RagB lid were analyzed more intensively. In accordance with the 3D model of RagAB on NCBI-iCn3D and Goulas [26], the anchor/hinge region (Appendix A, left), connecting the lid with the outer membrane, starts with a lipidated cysteine (C20, highly conserved even for outgroups but not for SusD). This is followed by a domain of 17 amino acids (X21-X38) whose C-terminal region presumably forms the hinge/pivot point that allows the “rigid-body movement of RagB” [18]. There are two amino acids (L22 and R24) that are the same for all RagB variants as well as for SusD and therefore structurally most essential. From position X27-X38, a colorful picture (again reflecting W83 similarity, hydropathy, RasMol visualization of properties, and side chain size of aa) emerges, which suggests that the hinge is constructed differently for the four types, with a potential impact on the opening and closing movement. A clear pattern linking type to practice is difficult. Madej et al. further depicted a RagB region close to the substrate (X76 and X83, here slightly expanded to X75-X84 for larger insight) (see Appendix A for three-dimensional structure and Appendix A-middle for multi-sequence alignment). As D77 and G78 are conserved for all four RagB types, they might play an important role regarding substrate binding. For position X79, RagB-1, -3, and -4 possess a polar uncharged asparagine, while RagB-2 has a hydrophobic glycine. Another outlier is RagB-4 at position X83 with arginine (positively charged) instead of proline (hydrophobic) for RagB-1-3. However, in this case RagB-2 again shows the most differences, as the last shown amino acid (X84) is a tyrosine (polar) for type-2 instead of phenylalanine (hydrophobic). From a wider perspective, RagB-2 appears to be less hydrophobic in this region and especially in contrast to RagB-1, which is redder (=hydrophobic). As the substrate is absorbed solvent-free, the hydrophobicity could support absorption in RagB-1. In addition, a number of smaller side chains were identified in RagB-2, which could also affect substrate binding. Finally, the acidic loop insertion in RagB-1 (_99_DEDE_102_) and—in altering form—in RagB-3 (_99_DED_101_) mentioned by Madej et al. [18] was studied in detail (Appendix A, right). Interestingly, our amino acid analysis showed that the RagB-1 *P. gulae* strains differ from the human strains by one amino acid: instead of asparagine (D101), glycine (G101) was found. This could potentially indicate a gain of the acidic characteristic (with preference for alkaline substrates) by human *P. gingivalis* strains in the sense of an evolutionary event.

## 3. Discussion

### 3.1. Benefits and Limits of Universal RagA-Gene-Directed PCR Method

Because periodontitis is a widespread chronic inflammatory disease, not only affecting the oral cavity but also general health of patients [3], the importance of efficient diagnostic tools for the application of targeted therapeutic strategies is apparent. Research into the bacteria that cause periodontitis and in particular their ability to damage cells or ensure their own survival—as virulence factors do—is therefore essential. In contrast to other virulence factors like fimbrial adhesion proteins Fim/Mfa or gingipains, RagAB and the *ragAB* locus are not yet that well studied. Additionally, the four different *rag* alleles (with importance for virulence) make identification difficult.

As RagA is a transmembrane transporter, there are a whole series of homologous regions distributed across the entire genome. This hindered a direct *ragA* amplification as—even after testing several primer pairs—cross-reactions occurred, and presumptive *ragA* amplicons contained a mixture of transporter sequences. Instead, for the selective and type-independent sequencing of the *ragA* gene, an extended Long-PCR-product of 6–8.5 kb was amplified, including *ragA* (PG_0185) and *ragB* (PG_0186) as well as the *ragAB* genetic environment. This Long-PCR (LPCR) was introduced by Hall et al. in 2005, and for architecture of the different PCR products see Figure 2 of the same publication [16]. As IS elements can be found upstream of *ragA* and thus downstream of PG0183 (encoding for minor fimbriae ancillary tip subunit Mfa5), they might influence gene expression, an assumption that needs to be verified in future experiments. Of note here, the duplication of *mfa5*, found in a few strains [27], could be due to such IS elements. In fact, *rag-1* strains A7436 and HG66 both have an IS*Pg* element and show an *mfa5* duplication, with the *mfa5-2* variant being longer, while strains ATCC 33277 (*rag-4*) and TDC60 (*rag-4*) neither have IS*Pg* elements nor show an *mfa5* duplication. The LPCR products provided the baseline for the development of universal, type-independent *ragA* primers during our study and—by combining our five *ragA* with the eight *ragB* sequencing primers of Hall et al. [16]—both genes can be amplified and sequenced for any strain.

Limiting or aggravating factors are that double purification might be needed, as the smallest residuals of LPCR primers were found to interfere with sequencing. The principal reason could be secondary structures of the Long-PCR product impeding access of sequencing primers, a matter that has been discussed in previous publications [28]. In our experiments, this was necessary for the two reversed primers ragA-R4 and -R2, most likely due to their many wobbling bases. Additionally, the increase in sequencing primer’s concentration might help to reduce the impediment described above. Since ragA-R2 could not provide satisfactory results for type-4, ragA-R2a was added, which starts 80 bases further away from the *ragA* end but shows fewer wobbles and thus less cross annealing.

Nevertheless, this new way of detecting all types of *ragA* contributes to improving *P. gingivalis rag* locus detection and thus therapy-relevant information acquisition. Knowing the type of *ragAB* and the corresponding level of virulence can help determine the appropriate therapeutic approach, for example, by treating type-1 more aggressively with antibiotics such as metronidazole [29].

As genome-derived (mapped) sequences might have gaps (such as the *rag* locus of OMI 731 in our data set) or sequencing errors, the LPCR-*rag* amplification and sequencing approach we developed here can be used to resolve ambiguities. In addition, re-sequencing a single or a few bacterial genomes is very expensive and takes weeks or months to complete. So, the LPCR-*rag* sequencing approach here saves money and time.

In the future, the introduction of deoxy-inosine may allow type-independent *ragB* amplification by substituting the most wobbled positions in primer sequences, thus solving the problem of heterogeneity. Inosine primers have already demonstrated their advantages in the broad amplification and sequencing of the taxonomically important but highly heterogeneous *rpoB* gene, allowing a novel assay for broader bacterial identification in clinical microbiology [30].

### 3.2. New Insights into RagAB Phylogeny and Evolution

Hall et al. investigated the diversity of the *rag* locus and showed that there are at least four different variants (*rag-1* to *rag-4*) [16]. Since the *ragA* and *ragB* genes were found in all *P. gingivalis* strains investigated, we were able to confirm the universal presence of *rag* in *P. gingivalis*, but different types may be present at different frequencies. Our results differed slightly from the study of Hall et al. [16], where *rag-4* was 14%, *rag-1* was 26%, and *rag-3* was 25%, whereas we found about the same percentage (16–17%) for all three. However, their calculation for *rag-2* being the most common (36%) was confirmed by our data (almost 50%). In conclusion here, geographical differences in *rag*-type prevalence might exist, but these results need to be interpreted with caution.

Regarding the within-patient evolution of *P. gingivalis*, we confirm that most patients sampled twice over eight years (1994 and 2002) showed the same strain and *rag* type. However, in other cases different strains with the same or different *rag* types were isolated from the same patient. Taken together, this pattern of in-host evolution is similar to that described for *Stenotrophomonas maltophilia* genotypes in cystic fibrosis [31].

All of these evolutionary processes and adaptations can have a variety of possible causes, including not only natural selection for the fittest but also host-specific selective pressures including biological, environmental, and behavioral factors. In particular, our immune system may select for traits that enable the survival and persistence of certain strains or communities [32]. Furthermore, the host environment can not only provide different mobile genetic elements but also modulate both recombination rates and mutational signatures [33].

### 3.3. RagAB as Factors of Virulence and Fitness

Diard and Hardt discussed different definitions of “virulence”, one at the level of individual host–pathogen interactions, where virulence increases colonization, exploitation, and damage, and one at the level of co-evolution, where virulence increases the fitness of a pathogen in the population [34]. In both definitions, “survival” is central; thus, most virulence factors are no more than survival factors that stimulate growth in the host environment and ensure persistence [35]. As an asaccharolytic species, the growth and survival of *P. gingivalis* depends on protein uptake through RagAB, modified type-specifically. The variability in colonization time, which can range from a few days to more than a week, is a well-documented phenomenon among *P. gingivalis* researchers. From the study of Madej et al., it is known that W83-RagAB-1 grows well on even minimal medium (supplemented with vitamin K1/K3, L-cysteine, and hemin), but ATCC 33277-RagAB-4 needed much more time, a deficiency which could be complemented by just replacing gene *ragAB-4* with *ragAB-1* [18].

The *rag* locus, as well as other virulence genes, is independently (uncorrelated) horizontally transferred between strains of the same species or even between different species [11,13]. However, Hall et al. found a limited degree of correlation between the k-capsular serotypes k1–k6 (plus k0 as un-capsuled) as k3 and k5 correspond to Rag-3, k4 to Rag-1, and k0–2 to two different *rag* alleles [16]. Furthermore, combining earlier results about long fimbriae (FimA) from of our group with the results about Rag type here, a few strains seem to accumulate more virulent types of both genes as *ragAB*-1 and *fimA*-IV are found together in strains of very different origin such as OMI 629 (W83, Germany), OMI 1079/OMI 1101 (Indonesia), and OMI 1049 (USA) [8].

In the same study, it was found that the animal *P. gulae fimA* typeA is closely related to human *P. gingivalis* strains of cluster Ib, potentially representing an ancestor genotype. Since *ragAB* type-1, the most virulent, has been identified in *P. gulae* from monkeys, we propose here that it may represent the ancestral sequence of the *ragAB* gene cluster (designated *rag-1a*). However, according to our phylogenetic trees, several other animal lineages exist, and horizontal transfer of the *ragAB* gene cluster between *P. gulae* and *P. gingivalis* may be possible on several independent occasions and may even be ongoing, as suggested by Meyer et al. and Fujiwara-Takahashi et al. for the *fim* type [8,36].

Another aspect that can be discussed is the N-terminal highly conserved part of *ragA*. This N-terminus contained a 20 aa signal peptide (MKRMTLFFLCLLTSIGWAMA) and a carboxypeptidase D_ regulatory-like domain (CarbopepD_reg_2, PF13715) found in bacteria, archaea, and eukaryotes, of approximately 90 aa in length, also known as N-terminal extension (NTE)—as preceding the TonB box—but with an unknown function so far. However, homologues are known to be collagen-binding [37] and might have proteolytic activity. Three-dimensional structures of RagA (deduced from W83) suggest that this domain is not part of the mature protein, but this has to be confirmed.

Still, there are no molecular explanations on why *rag-1* leads to a higher virulence than the other three *rag* alleles, leading us to the next section.

### 3.4. Substrate Specificity and Binding/Uptake Capacity as Explanation for Virulence

Recent analysis on the structure of the RagAB protein complex proved that this transporter is responsible for the uptake of proteinaceous nutrients [18]. The exact substrate-binding sites can be revealed when sites of different substrates (glycans and proteins) are compared, as we carried out in this study here. Our protein alignment of RagA and SusC confirmed the hypothesis that for RagA a potential glycan-binding site is missing while different (type-specific) peptide-binding sites appear instead. In total, RagA interacts via minimal 26 aa residues and RagB via minimal 8 aa residues with the bound peptide [18]. This allows type- and peptide-specific variation in nutrient uptake. However, *ragA* type (barrel) and *ragB* type (lid) are corresponding and thus dependent, and both together define not only the substrate preferences but also the uptake capacity and speed. For instance, the sheer size of the cavity between lid and barrel could also be decisive for the preferred volume of substrate. According to Glenwright et al. [38], for BT2263 (SusD in *B. thetaiotaomicron*) and BT2264 (SusC), a deca-glycine or even several peptides can fit into this hollow space. With a contact area of ~3800 Å^2^ [38] and a volume of 11,500 Å^3^ [18], the calculated height of 3.0 Å results in only one layer of peptides. However, these assumptions have yet to be proven for RagAB.

Related to uptake speed, our hypothesis is that this may depend on how high the lid can open (with loop L7 as pedal) before loop L8 limits the range. In RagA-2, loop L7 contains more amino acids, and such a longer pedal may modulate the dynamics either by allowing a higher substrate capacity or by slowing down the uptake process compared to the other types. The latter might be more likely, as the substrate/ligand binding itself leads—like a zipper—to the bin closing, and this might take more time if the aperture angle between lid and barrel is larger. In conclusion here, the virulence might just be driven by *P. gingivalis* strain-dependent “appetite” and how important the preferred substrates (proteins) are for periodontal integrity and/or immunological defense.

For RagB, a meaningful transfer of SusD information—as performed above for RagA/SusC—was much more difficult. Corresponding to Pollet and coauthors addressing TBDT in *Bacteroides*, the C-terminal region containing the ligand-binding region is much more variable [39]. The length of the different SusD/RagB proteins varies fundamentally among different members of the SusD/RagB family, with most variables at the C-terminal region, making it even harder to draw conclusions on the substrate-binding site. The RagB/SusD variability itself could even be the principle for keeping the choice of food as flexible as possible. Finally, RagB could just be a rigid lid (or trap) that forms a cavity (of about 11,500 Å^3^) together with RagA and opens (by about 35 Å) while waiting for the ligand, a hypothesis that needs to be proven, of course. Nevertheless, the immunological function of RagB as a pro-inflammatory signal transducer as well as toll-like receptor 2 (TLR2) and TLR4 agonist [26,40] should not be disregarded.

According to new findings by White et al., additional outer membrane components (namely a surface glycan-binding protein, SGBP, and a corresponding glycoside hydrolase, GH) assemble on the core SusCD transporter in *Bacteroides*, forming a stable, octameric glycan-utilizing machine that they termed *utilisome* [23]. In the future, it will be exciting to discover if RagAB forms not only tetrameric structures but such octameric utilisomes. It is plausible that RagAB does not stand alone but that it is interacting with (a variety) of surface protein-binding molecules (SBPs), attracting the substrate and (a variety) of peptidases digesting it before. The only question is how distant these two other partners are, either floating in the outer membrane (such as earlier speculated for SusCD [18,41]) or directly attached to RagAB, as proven for SusCD recently [23]. In the latter model, there is need for binding of much bigger molecules such as SBPs and enzymes and thus for different lid-opening angles as we found some evidence for in the hinge/pivot region of our sequences. Nonetheless, confirmation needs sophisticated quantitative proteomics and single-particle cryo-EM investigations.

As a limitation of our study, it should be added that a direct comparison of SusCD and RagAB, even if performed critically, must be treated with caution. Although both TBDTs have a similar structure, they occur in different species with very different substrate preferences. We are very confident that a few key structures were identified on species and type levels. However, how essential these domains are can only be proven by knock-out mutations and/or animal trials.

In conclusion, despite some limitations, our analysis of the 129 *P. gingivalis*/*P. gulae* genomes from our own collection and the 100 best-matching WP sequences imported from NCBI enabled us to further explore the genetic and functional diversity of RagAB, encoded by four *rag* alleles. A way to detect all four *ragA* variants was realized by combining an already known Long-PCR with our new sequencing primers. In addition, a potential plug-opening mechanism for RagA in the form of the aromatic triplet was discovered through the transfer of SusC information. The essential TBDT functions are more dependent on the RagA barrel than on the RagB lid. In general, the complex seems to be quite flexible with respect to the substrate, with the exception of RagA-1, which probably indicates a “special diet”. In addition, an insertion (8 aa long) found in loop L7 throughout RagA-2 might not only affect the dynamics of lid opening/closing but also alter the associated substrate throughput rate.

## 4. Material and Methods

This study can be divided into two parts: In the first part, phylogenetic trees based on genome mapping and corresponding amino acid sequences of our collection, further expanded by GenBank (NCBI) entries, were constructed and domain-specifically analyzed. In the second part, an attempt was made to develop a new PCR method for universal *ragA* and *ragB* amplification and sequencing. Figure 3 represents a graphical overview of our study.

### 4.1. Bacterial Strains and Growth Conditions

The basis of this study was a *P. gingivalis* collection (Division of Oral Microbiology and Immunology (OMI)) at the RWTH Aachen University reflecting a wide variety of strains (human and animal hosts, geographical origin, capsule type, fimbriae type). Further information on the isolates is shown in Appendix A. It is noteworthy that some species closely related to *P. gingivalis*, such as *Porphyromonas gulae (P. gulae)*, *P. loveana*, *P. macacae,* or *P. somerae*, were also added to the mapping for broader and reliable insight into *ragAB* evolution. All isolates were stored in cryotubes at −72 °C. The samples were cultivated on tryptone soya blood agar (TSBA) or Brucella blood agar and put under anaerobic conditions via BD GasPak^TM^ Gas Generating System bags (<1% oxygen, ≥13% CO_2_) for 3–7 days at 37 °C.

### 4.2. Genomic DNA (gDNA) Extraction

After cultivation, colonies were suspended in 1 mL 0.9% NaCl, washed, and bacterial genomic DNA was extracted using the spin column method in accordance with the manufacturer’s instructions (QIAamp DNA Mini Kit; Qiagen; Venlo, The Netherlands). Spectrophotometrically, the DNA concentration and purity were evaluated (NanoVue Plus Spectrophotometer, GE Healthcare Europe GmbH, Freiburg, Germany). The gDNA concentration was between 25 ng/µL and 91 ng/µL, and the purity was in a regular range from 1.8 to 2.0 (260 nm/280 nm). The extracted gDNA was either subjected to the whole-genome sequencing (WGS), using the Illumina NovaSeq 6000 platform (San Diego, CA, USA) generating 250 base paired (bp) end reads or to a *ragA*-type independent (universal) PCR.

### 4.3. Comparative Genomics and Phylogeny of RagAB

Mapping with 129 isolates of *P. gingivalis*/*P. gulae* and a few related species of the collection was performed. The genomic DNA was subjected to whole-genome sequencing (WGS) using the Illumina NovaSeq 6000 platform (San Diego, CA, USA) generating 250 base paired (bp) end reads. Demultiplexing of all libraries and for all sequencing data was performed using Illumina bcl2fastq software v2.20, and reads with a final length of less than 20 bases were discarded. For mapping, FASTQ files were aligned against the *ragAB*-type-specific reference genes using the Snippy pipeline (Appendix A). If the genome sequence data of the *rag* locus were not complete, a universal PCR was used (see Section 4.4). After mapping, the strains were assigned to one of the four different *ragAB* types. Appendix A shows two partial alignments (1100 bp each) of representatives of all four types and in both *ragA* and *ragB*.

Analysis of *ragAB*/RagAB phylogeny and RagAB functional domains: Alignments for each type were created by Seaview, version 5.0.5 [43], and phylogenetic trees were generated on DNA (nucleic acids) as well as—after in silico translation—protein (amino acids) level using the neighbor-joining model with 100 bootstrap replications applying MEGA11, version 11.0.13 [44] (DNA level: Appendix A; protein-level: Appendix A). Furtherly, for each of the four Rag types, one representative was chosen to run a protein BLAST on NCBI database (Rag-1: strain W83 [RagA-1: AAQ65420.1; RagB-1: AAQ65421.1]; Rag-2: strain 11A [RagA-2: SJL25370.1; RagB-2: SJL25368.1]; Rag-3: strain SU60 [RagA-3: SJL29627.1; RagB-3: SJL29621.1]; Rag-4: strain ATCC33277 [RagA-4: AUR48986.1; RagB-4: AUR49279.1]). Comprehensive alignments including the first 100 protein BLAST results for RagA and RagB covered all four types, and the related phylogenetic trees were calculated (phylogenetic trees shown in Appendix A) (MEGA11). Moreover, additional alignments with functional domains (RagA: TonB box, plug region, hinge-associated loops 7 and 8, substrate-binding sites; RagB: anchor/hinge, substrate-binding sites including X75-X84 and an acid loop), combining NCBI and our own strain data, were generated by the NCBI Multiple Sequence Alignment Viewer (1.25.0) (Appendix A). The three-dimensional crystal structure, deposed by White, Ranson, and van den Berg, and based on data of Madej et al. (2020) [18], which can be viewed at https://www.rcsb.org/structure/6SMQ (accessed on 30 November 2024) (W83, open/closed state), was also taken into account here.

### 4.4. Development of Universal RagA-Gene-Directed PCR Method

Since approaches for *ragB* universal amplification failed due to its high heterogeneity, the methods are limited to *ragA*. For the *ragA* universal PCR design, we used one representative for each *rag* type: type-1: *P. gingivalis* strain OMI 629, original name W83, isolated in Bonn, Germany, 1950s; type-2: OMI 1090, original name HW24D-2, isolated in Quebec, Canada, before 1993; type-3: OMI 1088, original name 59Pg1, isolated in Indonesia, 1994; type-4: OMI 1132, type strain ATCC 33277, isolated in USA, before 1981.

PCR amplification of *ragA*: For *ragA*, the Long-PCR protocol introduced by Hall et al. (2005) was used [16]. The ingredient composition and the cycling parameters were adjusted as follows. First, we created a mastermix of 49 µL for each sample that contained 34 µL nuclease-free water (Life Technologies Corporation, Austin, TX, USA), 10 µL of 5X PCR buffer (2 mM MgCl_2_; Roche Diagnostik GmbH; Mannheim, Germany), 3.0 µL of nucleoside triphosphate set (dATP, dCTP, dGTP, dTTP; Roche Diagnostics GmbH; Mannheim, Germany), 0.5 µL of forward primer, 0.5 µL of reverse primer (each 1 µM, synthesized by TIB Molbiol Syntheselabor GmbH, Berlin; for sequences see Table 1), and 2.0 µL of LONG-Taq-DNA-Polymerase (New England Biolabs, Frankfurt am Main, Germany). Second, the mastermix was combined with 1 µL of undiluted gDNA. The cycling parameters used were the following: initial denaturation at 95 °C for 5 min; 25 cycles; per cycle denaturation at 95 °C for 1 min, primer annealing at 60 °C for 30 s, and for elongation 68 °C for 8 min; final elongation at 72 °C for 10 min. A positive control (DNA of reference strains) and negative control (water without any DNA) were always included. MicroAmp™ reaction tubes (Applied Biosystems, Waltham, MA, USA) were used, and the reaction was performed in a PCR Express Thermal Cycler (Thermo Hybaid, Ashford, UK). Via agarose gel electrophoresis, the Long-PCR products were visualized. Therefore, 1% gels were used by heating 0.6 g of agarose (Top Vision Agarose, Thermo Fisher Scientific Inc., Waltham, MA, USA) and 60 mL of TAE buffer (TRIS-Acetate-EDTA-buffer, SERVA Electrophoresis GmbH, Heidelberg, Germany) until the agarose was completely dissolved. As fluorescent colorant, 3.5 µL of Midori Green Advance (NIPPON Genetics EUROPE, Düren, Germany) was added. A volume of 8 µL of LPCR products was mixed with 3 µL of Blue Marker (40% saccharose and 0.1% Bromophenol blue) and pipetted in the gel pockets. For every run, a MassRuler High Range DNA Ladder (Thermo Fisher Scientific Inc., Waltham, MA, USA) was subjoined. The parameters for electrophoresis were 100 V and 250 mA for 60–90 min. With the help of UV light (GelStudio SA, Analytik Jena GmbH, Jena, Deutschland), the bands became visible. The Long-PCR products were purified in accordance with the manufacturer’s instructions (NucleoSpin^®^ Gel and PCR Clean-up, Macherey + Nagel, Düren, Germany). The purified product was sent for Sanger sequencing (Eurofins Genomics, Ebersberg, Germany). For this sequencing reaction, 5 µL of each LPCR product (20–80 ng/µL) and 5 µL of the newly designed five *ragA*-type-independent primers (ragA-F0 and -F1 each 5 µM, all others 20 µM) were mixed (Table 1). The *ragA* primers were synthesized by TIB Molbiol. The sequencing results were aligned/merged by the ClustalO algorithm using Seaview [43] to obtain an almost complete and ambiguity-free sequence for every type (allele). Figure 4 visualizes the LPCR product, the *ragAB* loci, the coverage of ragA primers, and flanking genes of *P. gingivalis* strain OMI 629 (W83, type *ragA-1*) as an example. In Appendix A, representative results of the other *ragA* types are shown, namely type-2: OMI 1090 (HW24D-2), type-3: OMI 1088 (59Pg1), and type-4: OMI 1132 (ATCC 33277).

## Figures and Tables

**Figure 1 ijms-26-01073-f001:**
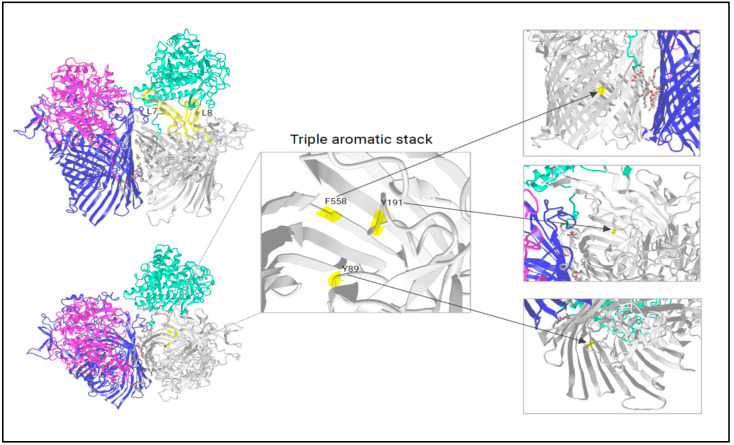
RagAB functional domains during substrate transport: loops L7 and L8 are highlighted in yellow in image **upper left**. Comparing the conformation (movement) in the open and closed states, L7 appears to be important for pedaling, while L8 is important for lid holding. At the **bottom left**, an overview of the position of the triple aromatic stack is given with a zoomed image as well as each participant from a different perspective: F558 from the outside and at the back of the barrel, Y191 from the inside as part of the plug loop, and Y89 from below following the TonB box. Modified from iCn3D and created in BioRender [24].

**Figure 2 ijms-26-01073-f002:**
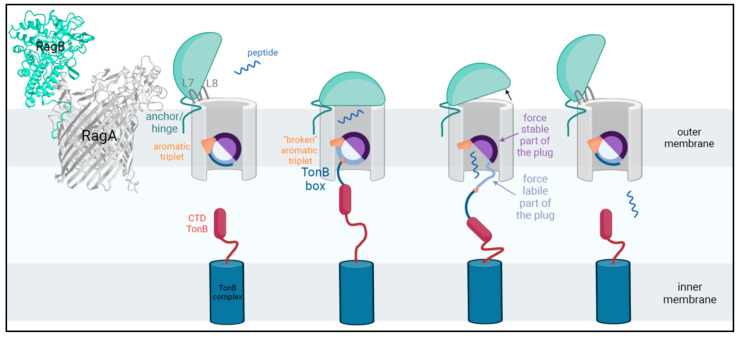
TonB-dependent peptide-substrate transport in *Porphyromonas gingivalis*. Only in substrate-loaded status, when the aromatic triple is disrupted, is the shifting TonB box able to interact with TonB-CTD, located in the inner membrane. TonB transduces the energy stored in the proton gradient to exert a force on the mechanically labile subdomain (N-terminal, light blue) in contrast to the solid domain (C-terminal, purple). Once the plug has been removed from the β-barrel lumen, substrates are able to diffuse to the periplasmatic space and subsequently gain access to the cytoplasm via ABC transporters. Modified after Madej et al. 2020 and White et al. 2023 [18,23] and created in BioRender [25].

**Figure 3 ijms-26-01073-f003:**
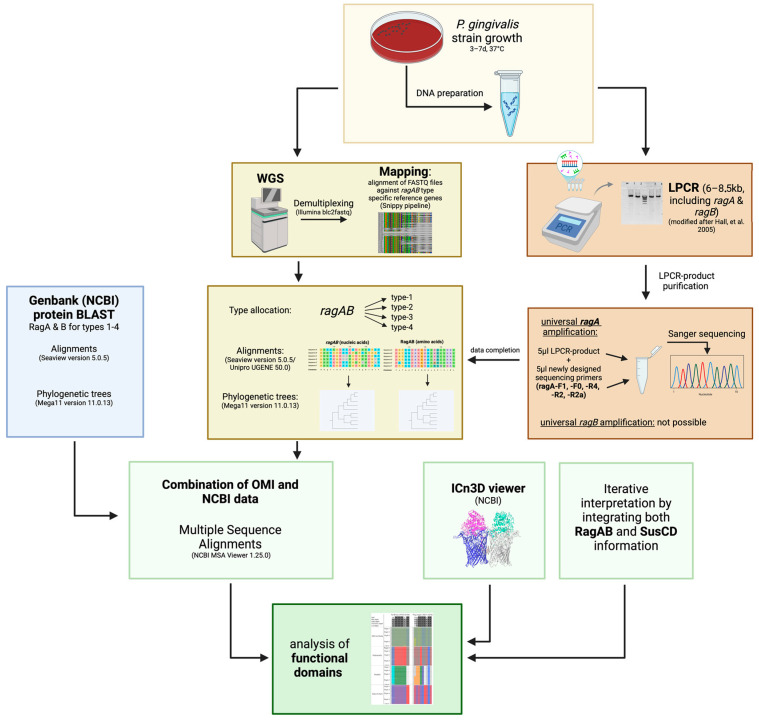
Graphical abstract of methodological flow of study. Abbreviation explanation: Long-PCR (LPCR), whole-genome sequencing (WGS), Division of Oral Microbiology and Immunology (OMI). Created in BioRender [16,42].

**Figure 4 ijms-26-01073-f004:**
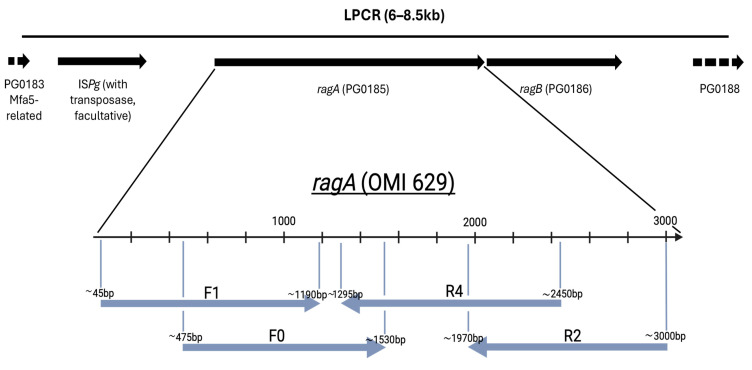
By the Long-PCR, the *rag* locus was amplified with flanking regions such as IS*Pg* transposase (facultative), partial PG0183 (3′ end with only 48 bp, encoding for minor fimbriae subunit Mfa5), and partial PG0188 (5′ end with 408 bp, encoding for a conserved protein of unknown function containing a sialidase-like Asp-box; [45]). With the double-cleaned Long-PCR product, *ragA* could be sequenced type-independently and, subsequently, the *ragB* type-dependently. Note that this approach does not cover the first and last 50 bp of *ragA*.

**Table 1 ijms-26-01073-t001:** Primers used for Long-PCR amplification and subsequent sequencing to cover all four *ragA* alleles (*ragA-1* to *ragA-4*). Primer ragA-R2a was applied in case ragA-R2 did not generate any sequences, especially occurring in type-4 strains. R: purine G or A; Y: pyrimidine C/T; M: amino A/C; W: weak A/T; N: all nucleotides A/T/G/C.

	Primer	Sequence	Length
PCR	LPCR-F	5′ CAA AGT CCT GCC ACG AGT AGC 3′	6–8.5 kb
primer	LPCR-R	5′ CGT TTT CTC GCC ACT TTC GTC 3′	
			Position Start/End
Sequencing	ragA-F1	5′ ATG AAA AGA ATG ACG CTA TTC TTC C 3′	50/1170
Primer	ragA-F0	5′ GGT CAG GTA GCC GGT ATG CAG GTT AT 3′	480/1540
	ragA-R4	5′ CCR GGR ACA TAC CAC A 3′	2500/1370
	ragA-R2	5′ TTA RAA MGA MAN YTG RAT ACC 3′	3030/2000
	ragA-R2a	5′ GGG TCR AAR CCT TTR WAC TT 3′	2980/1900 (type-4)

## Data Availability

The data that support the findings of this study are available from G.C. upon request. Genome data of *ragAB* were deposited by NCBI-BankIt under Accession-Nos. PQ658417-PQ658670.

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
