# Peer review of "Exploring the Genetic and Functional Diversity of Porphyromonas gingivalis Survival Factor RagAB"

_ijms, 2025, doi:10.3390/ijms26031073_

Round 1

Reviewer 1 Report

Comments and Suggestions for Authors

Exploring the genetic and functional diversity of Porphyromonas gingivalis survival factor RagAB

Montz, PG; Dafni, E; Neumann, B; Deng, D; Abdelbary, MMH; Conrads, G

General comments

The periodontal pathogen Porphyromonas gingivalis is known to co-express immunodominant antigens, RagA and RagB, that exhibit homology to SusC and SusD respectively, to Bacteriodes thetaiotaomicron sugar utilisation TonB-related transporters. Although there is only one ragA-ragB locus in P. gingivalis strains, there are 4 alleles that are highly heterogeneous and imparts different degrees of virulence potential to the bacterium; the locus has been shown to transport oligopeptides that are essential for growth.

Here, Montz and co-investigators have expanded on their earlier report on strain heterogeneity, in detection and DNA sequencing, to explain the extreme phenomena using molecular tools that are essentially summarised by Figure 3, to include related species; developing additional sets of primers, and sequencing amplicons / genomes for the loci in 129 Porphyromonas isolates.

Specific comments

Line 141. Serotypes is generally reserved for capsular polysaccharide.

Line 251. “AND”?

Line 374. “…fimbriae…” or “…fimbrillin…”? Also, legend to Figure 4.

Lines 387/ 388. Italise ragA.

Line 499. “…angles…” instead of “…angels…”?

Line 587. Check year of isolation of P. gingivalis W83.

Line 595. Check MgCl2 concentration. It appears low.

Section 3.3. Laboratory cultivation of P. gingivalis ATCC 33277 is more effective in the presence of menadione (vitamin K) in contrast to P. gingivalis W83. Is there mileage in here?

Tables 1.

Abbreviations for nucleotide redundancies?

Figure 3.

Move agar plate/annotation above DNA preparation.

Figure 4.

Is ISpg ISpg1? Check spelling in brackets.

Legend to Figure 4. “…facultative…”?

Author Response

Rev. 1: General comments

The periodontal pathogen Porphyromonas gingivalis is known to co-express immunodominant antigens, RagA and RagB, that exhibit homology to SusC and SusD respectively, to Bacteriodes thetaiotaomicron sugar utilisation TonB-related transporters. Although there is only one ragA-ragB locus in P. gingivalis strains, there are 4 alleles that are highly heterogeneous and imparts different degrees of virulence potential to the bacterium; the locus has been shown to transport oligopeptides that are essential for growth.                                                                                                                                              Here, Montz and co-investigators have expanded on their earlier report on strain heterogeneity, in detection and DNA sequencing, to explain the extreme phenomena using molecular tools that are essentially summarised by Figure 3, to include related species; developing additional sets of primers, and sequencing amplicons / genomes for the loci in 129 Porphyromonas isolates.

Answer: Thank you very much for summarizing the intention of our work so clearly.

Specific comments

Rev. 1: Line 141. Serotypes is generally reserved for capsular polysaccharide.

Answer:  Thank you for the comment. We would like to slightly disagree on this point, as serotypes can not only be capsule proteins (K-antigens such as k1-k6 of P.g.), but also other immunogenic epitopes such as flagella (H-) or the cell wall (O-) antigens.

Rev. 1: Line 251. “AND”?

Answer: Thank you; we now avoid the capital letters form. See change in text.

Rev. 1: Line 374. “…fimbriae…” or “…fimbrillin…”? Also, legend to Figure 4.

Answer: Thank you for questioning the term “minor fimbrium ancillary tip subunit Mfa5”. We copied this from the corresponding annotation without questioning. You are right; the singular of fimbriae is not “fimbrium” but fimbria; the plural you recommend is even better as much more in use; we changed to fimbriae (see text).

In the legend, we corrected the same mistake also.

Rev. 1: Lines 387/ 388. Italise ragA.

Answer: Thanks for your comment but in this case ragA-R2 and ragA-R2a are the primers, not the genes, and thus not italised.

Rev. 1: Line 499. “…angles…” instead of “…angels…”?

Answer: Thanks a lot for the comment, of course it should be “angles”, not “angels” (matter of auto-correction we guess but easy to overlook). The corresponding position is updated in line 522-revised version.

Rev. 1: Line 587. Check year of isolation of P. gingivalis W83.

Answer: Thank you very much for this hint. Great that you are familiar with such details. We checked the isolation year again and found the 1950s as decade of isolation (mentioned in Loos et al. 1993: “strain W83 was isolated in the 1950s by H. Werner from an undocumented human oral infection and was brought to The Pasteur Institute by Madeleine Sebald during the 1960s”).

The corresponding region in the text (line 614 revised version) is changed.

Rev. 1: Line 595. Check MgCl2 concentration. It appears low.

Answer: Thank you very much for this note and your sharp eye. This must have been a typing error, of course 2 nM is wrong, it should be 2 mM. [The position in line 621 revised version was updated]

Rev. 1: Section 3.3. Laboratory cultivation of P. gingivalis ATCC 33277 is more effective in the presence of menadione (vitamin K) in contrast to P. gingivalis W83. Is there mileage in here?

Answer: Thank you for pointing this out. I guess with the phrase “Is there mileage in here” you mean if menadione (vitamin K3) might also be a factor creating distance (mileage) between different strains with different growth profiles. In their studies, Madej et al. used 0.5 mg menadione as well for general growth conditions but also in minimal medium. We agree that nutritional supplements (with vitamin K derivates in the first instance) and their influence on growth dynamics should be more taken into account in the future. Regarding menadione a recent study (Saiki et al 2023, Journal of Oral Biosciences 65 (2023) 273e279) found that the situation (growth or inhibition) is even more complicate than previously thought and that there is even a difference between different ATCC 33277 stocks and between addition of vitamin K3 or K1 (phylloquinone). For instance they write “Menadione at 2.9 mM, the conventionally used concentration for culturing P. gingivalis, supported the growth of most ATCC 33277 strains [from different stocks they tested] but inhibited the growth of some W83 and ATCC 33277 strains. ”               

Taken together, the situation seems to be too complex to discuss in our RagAB-paper but it is interesting information for future studies indeed. Anyway, we added the sentence in line 443 “(supplemented with vitamin K1/K3, L-cysteine and hemin)

Rev. 1: Tables 1. Abbreviations for nucleotide redundancies?

Answer: Thank you for your question. I guess you mean that explanation of wobble positions are missing. We added to the legend: R: puRine G or A, y: pYrimidine C or T, M: aMino A or C, W: Weak A or T, N: any Nucleotide A, T, G, C

Rev. 1: Figure 3. Move agar plate/annotation above DNA preparation.

Answer: Thank you for this comment. We re-arranged the top-image with corresponding text to improve the logical order of work-flow.

Rev. 1: Figure 4. Is ISpg ISpg1? Check spelling in brackets.

Answer: This indeed was a discrepancy we also had to deal with. Background: according to the PROKKA Annotation, it’s ISPg8, but NCBI annotate it as ISPg1. As consequence, we simply named it “ISPg.”

Regarding to the typo in brackets, we corrected “facultative”; see Figure 4new; thank you.

Rev. 1:  Legend to Figure 4. “…facultative…”?

Answer: Thank you for the question. In A7436 and HG66 (and W83) the ISPg was described but it is absent for ATCC33277, explaining the different Long-PCR product lengths. In the text we write “In fact, rag-1-strains A7436 and HG66 both have an ISPg element and show an mfa5 du-plication, with the mfa5-2 variant being longer, while strains ATCC 33277 (rag-4) and TDC60 (rag-4) neither have ISPg elements nor show an mfa5 duplication.”

Taken together, the presence of an ISPg in the Long-PCR product is facultative.

We would like to thank the reviewer for his /her sharp eye for little flaws and enhancing our work.

Reviewer 2 Report

Comments and Suggestions for Authors

Dear authors, 

This manuscript addresses a valuable topic in molecular microbiology by exploring the genetic and functional diversity of Porphiromonas gingivalis survival factor RagAB. The study is well structured, with contributions highlighted by developing a universal PCR method capable of amplifying all rag variants, demonstrating the presence of the rag locus in a maximum number of strains, and classifying them accordingly.

The manuscript is well structured, with an introduction that provides detailed information on the topic, and the Materials and Methods section presents detailed information and provides the possibility of reproducing the study. From the point of view of the results, these are clearly presented with strong points: Phylogenetic Tree Building and the Analysis of RagA Functional Domains, which provide valuable insights into the molecular mechanism of RagAB. However, several aspects can be improved to increase the scientific impact of the paper.

Some speculative statements in the discussion section appear to have insufficient scientific support. I suggest they be supported by additional experiments or presented as hypotheses for future research.

Including a discussion of how ragB could be investigated in the future using other advanced sequencing methods is recommended.

Discussions that could link rag diversity to interactions with other genes or loci involved in P. gingivalis virulence provide a more comprehensive picture.

The discussion should better analyze the clinical implications of rag diversity.

Although genetic diversity is linked to adaptation, whether this is the result of natural selection or other selective pressures (e.g., host-specific immune responses) is not discussed.

Author Response

Revision to comments of Reviewer 2:

Rev. 2: Comments and Suggestions for Authors

Dear authors,

This manuscript addresses a valuable topic in molecular microbiology by exploring the genetic and functional diversity of Porphyromonas gingivalis survival factor RagAB. The study is well structured, with contributions highlighted by developing a universal PCR method capable of amplifying all rag variants, demonstrating the presence of the rag locus in a maximum number of strains, and classifying them accordingly.

The manuscript is well structured, with an introduction that provides detailed information on the topic, and the Materials and Methods section presents detailed information and provides the possibility of reproducing the study. From the point of view of the results, these are clearly presented with strong points: Phylogenetic Tree Building and the Analysis of RagA Functional Domains, which provide valuable insights into the molecular mechanism of RagAB.

Answer: Thank you very much for acknowledging our work.

Rev. 2: However, several aspects can be improved to increase the scientific impact of the paper.

Some speculative statements in the discussion section appear to have insufficient scientific support. I suggest they be supported by additional experiments or presented as hypotheses for future research.

Answer: We are very thankful for this criticism as it gives us the opportunity to improving our discussion prior publication. Indeed, we identified three statements where we added “hypothesis/need for future investigations” and another three  statements where we added more references, including one new one (Snyder et al 2008) about the influence on secondary structures on PCR performance, which can be too high – as in our case- but also too low (!)

Line 375 (IS-elements and gene-expression): an assumption that needs to be verified in future experiments.

Line 387 (impeding access of primers to secondary structures): a matter that has been discussed in previous publications [26].

Line 448: two references added.

Line 481: reference added.

Line 488 (size of peptide pocket): However, these assumptions have yet to be proven for RagAB.

Line 507 (RagAB as peptide trap): a hypothesis that needs to be proven, of course.

Rev. 2: Including a discussion of how ragB could be investigated in the future using other advanced sequencing methods is recommended.

Answer: Thank you for this helpful comment. We agree that attempts to develop a universal (type independent) ragB amplification were unsuccessful due to its high degree of heterogeneity (see lines 96-97. As we performed WGS and mapping we could derive the ragB information easily form this “left arm” of our study (see Figure 3). However, your question stimulated us to look for an alternative PCR-approach (right arm of our study) supporting the WGS and we found the application of inosine-primers (reducing wobbled positions) as probably useful for future research. [A corresponding passage and new reference has been added to the text in lines 403-408]

“In the future, the introduction of deoxy-inosine may allow type-independent ragB amplification by substituting the most wobbled positions in primer sequences, thus solving the problem of heterogeneity. Inosine primers have already demonstrated their advantages in the broad amplification and sequencing of the taxonomically important but highly heterogeneous rpoB gene, allowing a novel assay for broader bacterial identification in clinical microbiology (Bivand et al. 2024)”

Rev. 2: Discussions that could link rag diversity to interactions with other genes or loci involved in P. gingivalis virulence provide a more comprehensive picture.

Answer: Thank you for your objection. In our discussion we have already mentioned the utilisome and its SusCD -peptidases-SGBP-interactions which might be similar for RagAB-peptidases-SGBP also, but yet to discover (lines 514ff). Until now, however, no conclusions can be drawn about the RagAB-type-specific networking with surface proteins of other loci. Furthermore, k-capsule serotypes and fimA genes were linked to different rag types, an issue we already discuss (lines 448ff).                      Anyway, you are absolutely right that networking of virulence genes is a hot topic for future investigations.

Rev. 2: The discussion should better analyze the clinical implications of rag diversity.

Answer: Thanks a lot for this important remark. We agree that more emphasis should be placed on this aspect. On the one hand we already mentioned the importance of efficient diagnostic tools for the application of targeted therapeutic strategies (lines 359ff “the importance of efficient diagnostic tools for the application of targeted therapeutic strategies is apparent”, lines 394 “this new way of detecting all types of ragA contributes to improving P. gingivalis rag locus detection and thus therapy-relevant information acquisition”). With other words, knowing the type of ragAB and the corresponding level of virulence can help determine the appropriate therapeutic approach, for example, treating type-1 more aggressively with antibiotics such as metronidazole [Conrads et al. 2021]. [The highlighted text plus new reference was inserted in line 396 revised version]

Rev. 2: Although genetic diversity is linked to adaptation, whether this is the result of natural selection or other selective pressures (e.g., host-specific immune responses) is not discussed.

Answer: We agree that this is indeed an interesting and exciting aspect that needs further investigation. How far and in which way we as super-host influence microbial evolution should further elaborate on the question of genetic diversity. For instance, genomic recombination and mutation rates are driven by the niche, e.g. oral versus intestinal. Actually, we just published about the within-host evolution of streptococci, if you are interested (Int J Mol Sci. 2024 Dec 17;25(24):13507. doi: 10.3390/ijms252413507.) and we now cite under Abdelbary et al. 2024. The following text was added:

All of these evolutionary processes and adaptations can have a variety of possible causes, including not only natural selection for the fittest but also host-specific selective pressures including biological, environmental and behavioral factors. In particular, our immune system may select for traits that enable the survival and persistence of certain strains or communities [30]. Furthermore the host environment can not only provide different mobile genetic elements but also modulate both recombination rates and mutational signatures [31]. [This paragraph has been added in line 424ff]

We would like to thank the reviewer for his/her suggestions to improve our discussion substantially.
